# Efficacy of the Simeox^®^ Airway Clearance Technology in the Homecare Treatment of Children with Clinically Stable Cystic Fibrosis: A Randomized Controlled Trial

**DOI:** 10.3390/children10020204

**Published:** 2023-01-23

**Authors:** Dorota Sands, Katarzyna Walicka-Serzysko, Justyna Milczewska, Magdalena Postek, Natalia Jeneralska, Aleksandra Cichocka, Ewa Siedlecka, Urszula Borawska-Kowalczyk, Laurent Morin

**Affiliations:** 1Cystic Fibrosis Department, Institute of Mother and Child, 01-211 Warsaw, Poland; 2Cystic Fibrosis Centre, Pediatric Hospital, 05-092 Dziekanow Lesny, Poland; 3PhysioAssist, 34090 Montpellier, France

**Keywords:** cystic fibrosis, chest physiotherapy, homecare therapy, pulmonary function test, Simeox

## Abstract

Background: Cystic fibrosis (CF) patients require regular airway clearance therapy (ACT). The aim of this study was to evaluate homecare therapeutic effects of a new ACT (Simeox^®^) added to the optimal standard of care, including home chest physiotherapy, in the treatment of clinically stable children. Methods: Forty pediatric CF patients (8–17 years old) with stable disease were randomized 1:1 in a single-center, prospective, open-label, cross-over trial into two groups: with or without Simeox^®^. Lung function (impulse oscillometry, spirometry, body plethysmography, multi-breath nitrogen washout) results, health-related quality of life, and safety were assessed during the study after 1 month of therapy at home. Results: A significant decrease in proximal airway obstruction (as supported by improvement in airway resistance at 20 Hz (R20Hz) and maximum expiratory flow at 75% of FVC (MEF75)) compared to the control group was observed after 1 month of therapy with the device. Lung-clearance index was stable in the study group, while it worsened in the control group. In addition, the device group demonstrated a significant increase in the Cystic Fibrosis Questionnaire—Revised (CFQ-R) physical score. No side effects were identified during the study. Conclusions: Simeox^®^ may improve drainage of the airways in children with clinically stable CF and could be an option in chronic treatment of the disease.

## 1. Introduction

Cystic fibrosis (CF) is a genetic disease characterized by dehydration of airway surface liquid and impaired mucociliary clearance [1]. As a result, the lungs have difficulty eliminating pathogens, and patients suffer from chronic lung infections and inflammation [2]. The prognosis and quality of life in CF are determined by the course of bronchopulmonary disease. Nowadays, respiratory failure is still the most frequent cause of mortality related to CF [3]. It is mainly influenced by the development of infection with typical pathogens such as *Staphylococcus aureus*, *Pseudomonas aeruginosa*, fungi, and other pathogen-inducing factors and the intensity of daily regimens slowing down or even stopping its course.

The management of bronchial secretions is one of the main problems encountered in CF. Chest physiotherapy (CPT) is recommended to mobilize and remove airway secretions as a necessary therapy for people with CF [2,4,5]. Many different airway-clearance therapies (ACTs) have been developed and modified in order to optimize outcomes in CF [6]. Conventional therapy techniques typically consist of techniques such as modified postural drainage [7] and percussion and manual vibrations [8]. Although chest physiotherapy is widely prescribed to assist the clearance of airway secretions, airway-clearance therapies (ACTs) are currently being developed to reduce the harmful effects of airway obstruction in the lung parenchyma in CF. Despite various studies and systematic reviews having been carried out, the best technique has not yet been identified [9,10].

Based on the UK CF Registry data analysis regarding people with CF aged ≥11 years, 89% of 6372 patients were using ACTs [11]. The most commonly used techniques were forced expiratory techniques (28%) and oscillating positive expiratory pressure (O-PEP) (23%). In Bradley’s publication, there was a lack of evidence supporting long-term efficacy of airway clearance and physical training, and there is no evidence to support the substitution of airway-clearance sessions with physical training [12]. Since then, other Cochrane Reviews on airway-clearance therapies have been published [13,14]. The review [9] concluded that there is little evidence to support the use of one airway-clearance technique over another. Combining different mechanisms of action in CPT is a new field of investigation that is characterized by high clinical complexity that we will have to face [10]. With this in mind, newly presented airway-clearance technology (Simeox^®^, Physio-Assist) seems to be a viable option of CPT for patients with CF.

Positive expiratory pressure (PEP) devices, well-established in CPT, provide positive pressure to the airways during expiration [14]. It may improve clearance by placing air behind mucus via collateral ventilation and temporarily increasing functional residual capacity. To our knowledge, the presented device in this study is the first to generate exclusively short, high-frequency, negative pressure pulses during exhalation. Thanks to the intermittent oscillatory expiratory flow when the patient exhales freely, mechanical constraints applied on bronchi wall are very limited, and airways do not collapse. It leads to liquefying secretions and evacuating them from the respiratory system to be naturally expectorated.

Most people with CF have their own preferences for a particular airway-clearance technique, as shown by the drop outs in one randomized controlled trial (RCT) of long-term airway-clearance techniques [15]. Deciding which method will be used is often influenced by factors such as the patient’s age, symptoms, and needs. Selecting a method that best accounts for such factors may be challenging, especially given a lack of up-to-date published guidelines on the relative utility and reliability of the measures available.

The conventional CPT is performed in order to improve mucus clearance, to decrease the risk of pulmonary infection, slow the decline in pulmonary function, and improve quality of life (QoL) [2,4,5]. However, conventional CPT must be adjusted according to patient age, localization and quantity of mucus to expectorate, ease of use, and comfort. Indeed, it is a daily practice that requires time and effort from the patient.

The efficiency of airway clearance can be optimized with instrumental assistance. Safe, effective, and satisfactory self-administration are the pillars of any oscillatory device to be adapted as a means of routine airway-clearance intervention. To this day, however, none of the conventional chest physiotherapy techniques seem to be more effective than another, and none of the medical devices used has proven superiority in comparison with the conventional chest physiotherapy [2,16,17].

Simeox^®^ is a bronchial drainage device designed for patients with chronic respiratory diseases characterized by a large amount of secretion remaining in the bronchial tree, notably CF. The action of this device is based on the rheological and thixotropic properties of mucus. The device generates intermittent negative air pressure pulses of about 25 ms with a frequency of 12 Hz, which liquefies and mobilizes bronchial secretions in airways, allowing the patient to expectorate during free exhalation. A significant advantage of this method is that airways do not collapse during drainage. Moreover, the drainage is performed mainly in the distal parts of the bronchial tree, which is very important in obstructive respiratory diseases (in CF in particular), where the smallest caliber of the bronchi is most affected [10]. In our previous study, we showed trends to the lower lung-clearance index ratio as well as an improvement in the maximum expiratory flow at 25% of forced vital capacity in pediatric CF patients using the Simeox^®^ ACT device [18]. In the current study, we want to develop the results of our initial work providing the initial data on the use of Simeox^®^ in hospitalized CF patients.

The aim of this study was to assess the benefits and the safety of a new ACT (Simeox^®^, Physio-Assist) at home regarding pulmonary function and quality of life in children with CF.

## 2. Materials and Methods

### 2.1. Study Design and Ethical Considerations

The study is an open-label, prospective, monocentric, two-arm, cross-over randomized trial that was carried out at the CF Centre in Warsaw (Poland) from September 2019 to February 2021 (NCT04084041). The study was conducted according to the principals outlined in the Declaration of Helsinki and good clinical practice. The study protocol was approved by the local ethics committee (No52/2019). All patients and their legal guardians provided written informed consent before the enrolment in the study.

### 2.2. Patients

Patients aged 8–17 years with clinically stable CF who met the inclusion criteria and did not meet any exclusion criteria were eligible for the study and were invited to participate (Supplementary material S1).

Inclusion criteria included (i) diagnosis of cystic fibrosis based on current criteria [4,19,20], (ii) stable clinical condition (without pulmonary exacerbation four weeks prior to enrolment date), (iii) the ability to perform pulmonary function tests (spirometry, nitrogen multiple-breath washout (N_2_MBW), impulse oscillometry (IOS), and body plethysmography (BP)), and (iv) willingness to cooperate and learn the new technique of drainage.

Exclusion criteria included contraindications to CPT such as pneumothorax, hemoptysis, heart disease, recent chest injury or surgery, history of transplantation, or history of any other illness or any clinical condition that, in the opinion of the investigator, might confound the results of the study or pose an additional risk to the subjects.

### 2.3. Interventions

The included patients were randomly assigned to one of the two groups: group A (with Simeox^®^) and group B (without Simeox^®^; optimal standard of care (SoC), including home chest physiotherapy (CPT)). After 1 month of at-home intervention, the patients in group A switched to group B and vice versa. The study ended after another 1 month of at-home intervention (Figure 1).

CPT was performed three times a day: twice a day with Simeox^®^ with device settings individually adjusted for each patient by a respiratory physiotherapist. One session of Simeox^®^ consists of about 20–40 respiratory cycles and lasts about 20 min. Each respiratory cycle includes an inspiratory phase where the patient normally inspires and an expiratory phase where the patient expires in the mouthpiece while activating the device by pushing on the remote control. It is during the exhalation that the mucus is stimulated by oscillating successive negative pressure pulses and is liquefied. Once liquefied, the mucus can be easily expectorated by the patient. Morning treatment sessions included administration of bronchodilators, then nebulization of hypertonic saline, followed by autogenic drainage for 20 min (period of CPT without Simeox^®^) or autogenic drainage for 20 min with Simeox^®^ session (period of CPT using Simeox^®^). Afternoon sessions consisted of physical activity, then autogenic drainage for 20 min or autogenic drainage for 20 min with Simeox^®^ session, followed by nebulization with dornase alfa. Evening treatment sessions included bronchodilator administration, then nebulization of hypertonic saline, and O-PEP therapy with an individualized number of repetitions (drainage time 20 min) (Table 1)

### 2.4. Assessments

Medical history was obtained from hospital records. Patients performed spirometry, nitrogen multiple-breath washout (N_2_MBW), IOS, and BP at each of the study visits. At the same time, they completed the respiratory and physical functioning domain scores of the CFQ-R: a self-reported reliable and validated health-related quality of life measure that is specifically designed for patients with CF [21].

Spirometry was performed according to the American Thoracic Society/European Respiratory Society (ATS/ERS) criteria [22,23,24]. N_2_MBW was performed in order to calculate the lung-clearance index (LCI) [25]. N_2_MBW tests were performed and considered successful if there were at least two or more technically acceptable tests in accordance with guidelines in the ATS/ERS consensus statement [26]. In pursuance of the ATS/ERS criteria [23], BP was performed. Spirometry and flow-volume curves were measured. Results of three technically IOS acceptable measurements were used in the same mean resistance and reactance values [27].

### 2.5. Statistical Analyses

Demographic and Baseline Data

Descriptive statistics on demographic and baseline data were presented with mean and standard deviation for continuous variables and number and percent of patients for categorical variables. Lung-clearance index, spirometry, and body plethysmography data were transformed in z-scores to adjust for age, gender, height, and weight.

Data collected at baseline were compared between randomized groups using appropriate statistical tests (according to type of variable, normality of distribution, and verification of homoscedasticity) with R language. Qualitative data were compared using Fisher’s test or chi-square test, and quantitative variables were analyzed with Student’s *t*-test, Welch’s test, or Wilcoxon–Mann–Whitney test. (See supplementary data for more details.) The level of statistical significance was set below *p* < 0.05.

Patients in group A followed 4 days hospitalization with Simeox^®^ training, where lung function tests and quality-of-life questionnaire (CFQ-R) were performed. Patients then used Simeox^®^ at home for one month in addition to SoC. After one month, patients were present for a follow-up visit during which lung function tests were performed, and patients were instructed on the method of conventional chest physiotherapy (CPT). During the following month, patients in group A switched to group B and followed 1-day hospitalization (without Simeox^®^) with optimal SoC including conventional CPT. On the other hand, patients randomly assigned to group B followed an inversed procedure and were switched to the group A paradigm after one month. The study ended with a one-day hospitalization for patients in both groups A and B, during which PFTs were performed, and CFQ-R questionnaire was completed for respiratory and physical domain scores.

### 2.6. Comparison of Treatments

Linear mixed-effects model analyses were performed using R language (see packages in supplementary data Appendix A) to compare treatments effects on pulmonary function scores. Several models were built with different fixed effects ((1) group (A/B), treatment (Simeox^®^/conventional), and y at baseline or (2) group, treatment, y at baseline, and interaction between group and treatment) and a random subject effect (on intercept or on treatment slope). Then, the best model was selected according to the Akaike Information Criterion (AIC) [6].

### 2.7. Comparisons of Treatments Versus Baseline

Comparisons between treatments and baseline were conducted using linear mixed-effects models, with a similar approach as the one described above. Several models were built with different fixed effects ((1) group (A/B) and treatment (Simeox^®^/conventional/baseline) or (2) group, treatment, and interaction between group and treatment) and a random subject effect on the intercept. The best model was selected according to AIC. Validity of parametric models were verified, and transformations or non-parametric models were used when necessary. If the group effect or the interaction between group and treatment was significant, analyses were only conducted on the first series of values (visit 2).

## 3. Results

### 3.1. Baseline Characteristics of the Study Group

Over 14 months of the recruitment period (Sep 2019–Nov 2020), 40 patients with CF aged 8.12–17.60 years (mean 13.02 ± 2.80 years), with a total of 18 males, were enrolled in the studys. Their mean FEV1 was 90.88 ± 17.50%, mean FVC 97.40 ± 13.96%, mean LCI 2.5% 10.67 ± 3.68, and mean BMI 18.53 ± 2.47 kg/cm^2^ (Table 2).

### 3.2. Pulmonary Function Test Results

Pulmonary function test results showed some significant differences at baseline for the outcomes of FEV1, FVC, MEF at 25%, LCI2.5, VT(2), RV (%), and RV/TLC (Appendix A). This imbalance in baseline characteristics between groups was controlled thanks to the cross-over study design supporting a more effective evaluation of treatments than a parallel design.

The spirometry parameter, which showed significant trend toward improvement in the device group, was maximum expiratory flow at 75% of FVC (MEF75) and z-score 5.31 ± 1.04, while in the control group, it was 4.59 ± 1.04. Change in FEV1, FVC, and other spirometry parameters were similar in both groups. BP and IOS criteria did not change in both groups (Table 3).

The physical functioning score of the CFQ-R improved only in the Simeox^®^ group (90.4 ± 1.61) versus control group (87.0 ± 1.61; *p* = 0.015). The change in respiratory score was not different between groups (Table 4).

### 3.3. Mixed-Model Analysis Results

Additional analyses with mixed-model approach were performed by comparing each treatment against the baseline (Table 5).

As per the total lung resistance R 5 Hz, and according to the model results, when comparing baseline to optimal CPT treatment, the treatment effect was significant for the conventional treatment (*p*-value = 0.013). However, when comparing baseline to Simeox^®^ treatment, although the treatment effect was not significant for the R 5 Hz, the treatment effect showed a trend with a *p*-value = 0.067. In addition, the treatment effect was not significant for the R at 20 Hz (*p*-value = 0.103) when comparing baseline to optimal treatment. However, when comparing baseline to Simeox^®^ treatment, the treatment effect was significant for the R 20 Hz (*p*-value = 0.047).

When comparing the treatment effect of Simeox^®^, there was significant result for the MEF50 z-score with a *p*-value = 0.048. However, when comparing baseline to optimal treatment of the control group, the treatment effect was not significant for the MEF50, with a *p*-value = 0.503.

As the interaction between group and treatment effect was significant during the analysis of LCI2.5, a mixed model was re-built to compare treatments; group A baseline data were used for the comparison with the first series of Simeox^®^ values (visit 2), and group B baseline data were used for the comparison with the first series of conventional treatment values (visit 2). When comparing baseline to optimal treatment in the control group, the treatment effect was significant for the LCI2.5 before *p*-value correction for multiplicity (*p*-value = 0.014).

Interestingly, the treatment effect was not significant for the LCI2.5 (z-score) with a *p*-value = 0.365 when comparing baseline to Simeox^®^ treatment. However, when comparing baseline to optimal treatment in the control group, the treatment effect was significant for the LCI2.5, with a *p*-value = 0.007 (Table 5).

As per the CFQ-R physical score (patient), the treatment effect was not significant for the CFQ-R physical score (patient) for the conventional treatment versus baseline (*p*-value = 0.554). However, the results show an improvement for the Simeox^®^ group as compared to baseline, where the treatment effect was significant, with a *p*-value = 0.009 (Table 6).

### 3.4. Simeox^®^ Treatment Satisfactory Questionnaire

As shown In Figure 2, 95% of patients would recommend Simeox^®^ to other CF patients This is in line with the 72.5% of patients’ preference for drainage with Simeox^®^ over the existing drainage techniques. This high level of satisfaction could be explained by the autonomy of the patients to perform drainage without the help of a physiotherapist and level of comfort during the session. Additionally, the high level of satisfaction can be also reflected by the 77.5% reported absence of fatigue, 92.5% reported relaxed exhale, and 95% reported absence of drainage discomfort in the patients’ satisfactory questionnaire (Figure 2 and Supplementary material S6).

### 3.5. Adverse Events

Only one adverse event was reported in the study, and it occurred in one patient from the device group (Appendix A). The adverse event was a mild, not serious, hemoptysis not related to the device, which resolved without treatment.

## 4. Discussion

### 4.1. Spirometry, PFT, IOS, Multiple-Breath Washout, CFQ-R

R5Hz measured with IOS represents the total airway resistance, and R20Hz represents the resistance of the large airways, where R5-R20 reflects resistance in small airways. The current study results reflect changes in central resistance; thus, we suggest that airway obstruction in proximal airways was decreased in Simeox^®^. These results are in line with the increase of MEF75 with Simeox^®^ compared to optimal usual CPT. MEF75% is more sensitive than FEV1 as a measure of early or mild airway obstruction in children, as normal FEV1 is not indicative of normal spirometry. Thus, the current results show that Simeox^®^ has an effect on lung obstruction reduction. These results are in line with result published from the study by König and colleagues concluding that spirometry measures other than FEV1 should be considered in the clinical evaluation of airway obstruction [28].

Interestingly, the increase in the LCI2.5 observed in the control group showed a worsening of the ventilation inhomogeneity in children with CF and more specifically of the distal tract. This was not observed in the Simeox^®^ group, where LCI2.5 remained stable. Several studies have demonstrated evidence that LCI may be a potential outcome measure for interventional trials in patients with preserved lung function [29]. Furthermore, Gustafsson et al. found that tests of ventilation inhomogeneity such as the LCI, used in the current study, are more sensitive than spirometry in children with cystic fibrosis [30]. Thus. our results suggest that Simeox^®^ therapy could preserve lung function after 1 month of therapy. In the current study, the decrease in lung obstruction supports the stability of ventilation homogeneity. These results are similar to reported reduction in LCI following noninvasive ventilation [31].

The Cystic Fibrosis Questionnaire—Revised (CFQ-R) is a disease-specific health-related quality-of-life (HRQOL) measure for children, adolescents, and adults with cystic fibrosis (CF). As per the CFQ-R physical score (patient), the results also show significant results only with Simeox^®^. This adds important evidence to the efficacy of the device when compared to optimal standard care. The Simeox^®^ airway-clearance medical device used in the current study aims at minimizing the devastating effects of airway obstruction, infection, and inflammation due to mucus stasis on the conducting airways and lung parenchyma. In fact, the current results of amelioration of patients’ quality of life after using Simeox^®^ can be explained with a better airway clearance and thus better lung function, which plays a role in dictating physical activity improvements [11].

Of interest, all patients were highly satisfied by Simeox^®^ and stated that they could carry out their drainage with Simeox^®^ completely independently (without any help of a physiotherapist), which is extremely important in the era of the COVID-19 pandemic. Indeed, it was suggested that patients with cystic fibrosis may have a preference for self-administered treatments over convention chest physiotherapy [1,2,12,15].

### 4.2. Safety Results Discussion

There was only one mild, not serious, not related to the medical device, and not treated hemoptysis episode declared an adverse event. No other AEs were reported in this study; this is explained by the short-term study and the use of the autogenic drainage as manual chest physiotherapy technique in complement with incentive spirometry, which has a very strong safety profile [32]. In addition, there were no pulmonary exacerbations reported in this study (Appendix A).

### 4.3. Limitations

In this small study, no control group was included, and each patient served as its own control. Another limitation of the current study is the open-label design. Blinding aims to reduce investigator-related biases, and this was not guaranteed in the current study. Meanwhile, the study was performed in a controlled, randomized cross-over design, allowing more reliable study assessments. Regardless of the randomization, some outcomes were significantly different at baseline, and thus, for some outcomes, the balance was not ideal.

In cross-over studies, the study participants will be switched throughout to all the treatment groups (both test and reference formulations) after a washout period. In using the same set of the population, the advantage of cross-over studies is that patients act as their own controls. Additionally, the patients enrolled in the current study underwent randomization in order to eliminate the selection bias, balance the groups with respect to many known and unknown confounding or prognostic variables, and form the basis for statistical tests, which is a basis for an assumption of free statistical tests of the equality of treatments. Finally, one might discuss the short-term follow-up and low sample size of the current study as limitations challenging long-term randomized, real-world evidence. However, due to feasibility reasons—and even with unlimited funds—RCTs with longer follow-up paradigms can suffer dropouts and patient motivation-related biases [33]. More long-term, high-quality randomized controlled trials comparing Simeox^®^ to other airway-clearance techniques among patients with CF are needed in future studies.

## 5. Conclusions

People with cystic fibrosis should choose the ACT that best meets their needs after considering comfort, convenience, flexibility, practicality, cost, and other factors. This study results suggest that Simeox^®^ may improve safe drainage of the airways in children with clinically stable CF under optimal therapy and could be an option in the chronic treatment of CF. As per the benefits, the results of the current study suggest that Simeox^®^ may better manage CF in children in addition to the optimal standard of care by decreasing airway obstruction in proximal airways, maintaining lung-clearance homogeneity, and increasing the CFQ-R physical functioning, all with safety and a high level of patient satisfaction.

## Figures and Tables

**Figure 1 children-10-00204-f001:**
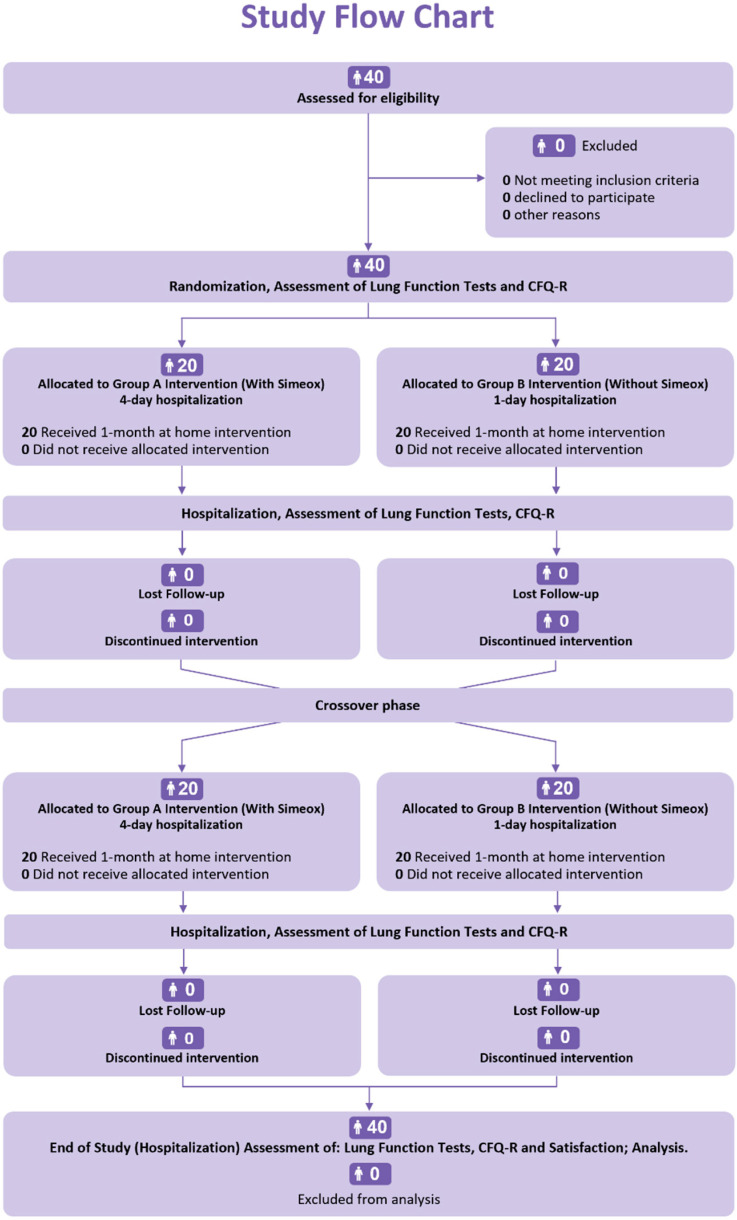
Timeline of the study.

**Figure 2 children-10-00204-f002:**
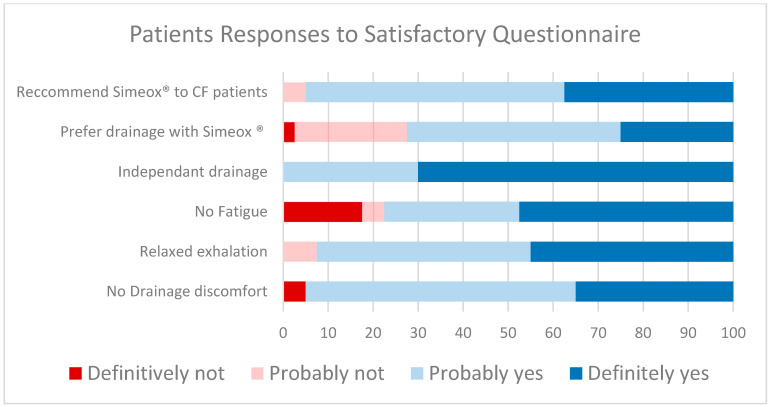
Patients Responses to Satisfactory Questionnaire.

**Table 1 children-10-00204-t001:** Planned care program in each group with and without Simeox^®^.

	Group Using the Simeox^®^ Technique	Group Using the CPT Technique
Morning	MDI or DPI bronchodilator	MDI or DPI bronchodilator
	Nebulization with physiological or hypertonic saline solution	Nebulization with physiological or hypertonic saline solution
	Autogenic drainage, drainage with Simeox^®^ technique for 20 min.	Autogenic drainage for 20 min.
Afternoon	Physical activity	Physical activity
	Autogenic drainage, drainage with Simeox^®^ technique for 20 min	Autogenic drainage for 20 min
	Nebulization with dornase alfa.	Nebulization with dornase alfa.
Evening	MDI or DPI bronchodilatorNebulization with physiological or hypertonic saline solutionDrainage with O-PEP (Aerobika, Flutter, Acapella) for 20 min.	MDI or DPI bronchodilator
		Nebulization with physiological or hypertonic saline solution
		Drainage with O-PEP (Aerobika, Flutter, Acapella) for 20 min.

**Table 2 children-10-00204-t002:** Demographic characteristics of the study group.

Demographic Characteristics	All Patients *n* = 40	Group A *n* = 20	Group B *n* = 20
Female, *n*	22	12	10
Male, *n*	18	8	10
Age of patient (years), Mean ± SD	13.02 ± 2.80	13.03 ± 3.05	13.01 ± 2.61
Height (cm), Mean ± SD	154.43 ± 13.87	152.40 ± 14.07	156.45 ± 13.72
Weight (kg), Mean ± SD	45.16 ± 12.52	42.89 ± 12.07	47.42 ± 12.86
BMI (kg/cm²), Mean ± SD	18.53 ± 2.47	18.06 ± 2.42	19.00 ± 2.49
Comorbidities			
Pancreatic insufficiency, *n* (%)	36 (90.0)	19 (95.0)	17 (85.0)
Diabetes, *n* (%)	15 (37.5)	9 (45.0)	6 (30.0)
Sinus polyposis, *n* (%)	26 (65.0)	14 (70.0)	12 (60.0)
Cirrhosis, *n* (%)	1 (2.5)	1 (5.0)	0 (0)
Chronic pseudomonas aeruginosa, *n* (%)	8 (20.0)	6 (30.0)	2 (10.0)

**Table 3 children-10-00204-t003:** Comparisons of treatment effects on pulmonary function tests (estimated mean after one month of intervention).

Criteria	Simeox^®^	Conventional	Simeox^®^—Conventional	*p*-Value
	Estimated adjusted Means ± SE or Medians (Q1; Q3)		Estimated Treatment Effect [CI95%]	Uncorrected/Corrected *^§^*
**Impulse Oscillometry**				
R 5 Hz	0.45 ± 0.01	0.44 ± 0.01	0.01 [−0.01; 0.03]	0.457/0.810
R 5–R 20 Hz	0.06 (0.03; 0.10)	0.06 (0.03; 0.09)	0.00	0.217/0.651
R 20 Hz	0.38 ± 0.01	0.38 ± 0.01	−0.00 [−0.02; 0.02]	0.721/0.824
X 5 Hz **	0.00 ± 0.01	0.03 ± 0.01	−0.03 [−0.06; 0.00]	0.060/0.394
AX	0.50 (0.34; 0.860)	0.45 (0.35; 0.76)	0.05	0.393/0.810
**Spirometry**				
FEV1 z-score	−0.57 (−1.25; 0.27)	−0.82 (−1.37; 0.37)	0.25	0.746/0.824
FVC z-score	−0.31 ± 0.08	−0.26 ± 0.08	−0.04 [−0.21; 0.19]	0.586/0.810
MEF25 z-score	−1.23 ± 0.13	−1.26 ± 0.13	0.03 [−0.23; 0.30]	0.785/0.825
MEF50 z-score	−0.53 (−1.38; 0.40)	−0.49 (−1.25; 1.12)	0.04	0.094/0.394
MEF75 z-score	5.31 ± 1.04	4.95 ± 1.04	0.36 [0.02; 0.13]	**0.008 ***/0.159
**Lung-clearance index**				
LCI 2.5 **	−0.37 ± 0.34	0.54 ± 0.34	−0.91 [−1.93; 0.11]	0.079/0.394
LCI 2.5 z-score **	−0.31 ± 0.57	0.93 ± 0.57	−1.24 [−2.97; 0.50]	0.156/0.545
**Body plethysmography**				
RV z-score	0.55 ± 0.26	0.39 ± 0.26	0.16 [−0.46; 0.77]	0.603/0.810
TLC z-score	−0.20 ± 0.17	−0.10 ± 0.17	−0.10 [−0.49; 0.29]	0.617/0.810
RV/TLC z-score **	0.09 ± 0.50	−0.48 ± 0.51	0.57 [−0.97; 2.10]	0.460/0.810
FRC z-score	0.64 (−0.66; 1.84)	0.32 (−0.85; 1.88)	0.32	0.519/0.810
Reff z-score	0.86 (0.01; 2.06)	0.79 (−0.39; 2.11)	0.07	0.524/0.810
sReff z-score	1.58 (0.87; 3.01)	1.55 (0.45; 2.90)	0.03	0.519/0.810
Rtot z-score	1.74 ± 0.09 *(in log scale)*	1.72 ± 0.09 *(in log scale)*	0.02 [−0.12; 0.17]	0.743/0.824

Legends: Forced expiratory volume in 1 s (FEV1), forced vital capacity (FVC), maximal expiratory flow (MEF), lung-clearance index (LCI2.5), airway resistance (sReff, kPa*s), airway resistance (Rtot, kPa/(l/s)), airway resistance (Reff, kPa/(l/s)), functional residual capacity (MBNW), total lung capacity (TLC), residual volume (RV), central lung resistance (R5hz), peripheral lung resistance (R5−20hz), peripheral lung reactance (X5hz), peripheral lung reactance (X5hz), area of reactance (AX), and resonant frequency (Fres). *p*-value in bold with * indicates *p* < 0.05; ** analyses were only conducted on the first series of values (visit 2) because there was a significant group effect or a significant interaction group x treatment effect when considering all data; *^§^* FDR correction.

**Table 4 children-10-00204-t004:** Comparisons of treatment effects on CFQ-R (estimated mean after one month of intervention).

Criteria	Simeox^®^	Conventional	Simeox^®^—Conventional	*p*-Value
	Estimated Adjusted Means ± SE or Medians (Q1; Q3)		Estimated Treatment Effect [CI95%]	Uncorrected/Corrected
**Cystic Fibrosis questionnaire (CFQ-R)**				
Physical score (patient)	90.4 ± 1.61	87.0 ± 1.61	3.37 [0.56; 6.17]	**0.015 ***/0.159
Respiratory score (patient)	82.7 ± 1.63	82.5 ± 1.63	0.21 [−3.04; 3.46]	0.897/0.897

Legends: *p*-value in bold with * indicates *p* < 0.05.

**Table 5 children-10-00204-t005:** Comparisons of pulmonary functions tests (treatments against baseline).

Criteria	Baseline **	Simeox^®^ Treatment		Conventional Treatment	
		After Treatment	Baseline–Treatment	After Treatment	Baseline–Treatment
	Adjusted Means ± SE or Medians (Q1; Q3)	Adjusted Means ± SE or Medians (Q1; Q3)	Treatment Effect [CI95%] Uncorrected/Corrected ^§^ *p*-Value	Adjusted Means ± SE or Medians (Q1; Q3)	Treatment Effect [CI95%] Uncorrected/Corrected ^§^ *p*-Value
**Impulse Oscillometry**					
R 5 Hz	0.48 ± 0.03	0.45 ± 0.02	0.03 [0.00; 0.05] -/0.067	0.44 ± 0.02	0.03 [0.00; 0.06] -/**0.013 ***
R 5- R 20 Hz	0.06 (0.03; 0.10)	0.06 (0.03; 0.10)	0.00 0.576/0.576	0.06 (0.03; 0.09)	0.00 0.138/0.275
R 20 Hz	0.40 ± 0.02	0.38 ± 0.02	0.02 [0.00; 0.04] -/**0.047 ***	0.38 ± 0.02	0.02 [0.00; 0.04] -/0.103
X 5 Hz	−0.17 (−0.20; −0.11)	−0.16 (−0.19; −0.12)	0.01 0.432/0.432	−0.15 (−0.19; −0.12)	0.02 **0.038 ***/0.077
AX	0.54 (0.30; 0.89)	0.50 (0.34; 0.86)	0.04 0.673/0.106	0.45 (0.35; 0.76)	0.09 0.053/0.135
**Spirometry**					
FEV1 z-score	Group A **: −1.16 (−2.66; −0.40)	−0.79 (−1.93; −0.24)	0.37 0.779/0.779	−0.55 (−0.91; 0.39)	−0.26 0.092/0.186
	Group B **: −0.29 (−0.45; 0.40)				
FVC z-score	−0.24 ± 0.19	−0.29 ± 0.20	0.05 [−0.20; 0.30] -/0.871	−0.33 ± 0.20	0.09 [−0.16; 0.34] -/0.657
MEF25 z-score	−1.08 ± 0.24	−1.23 ± 0.24	0.15 [−0.17; 0.46] -/0.503	−1.26 ± 0.24	0.18 [−0.13; 0.50] -/0.351
MEF50 z-score	−0.21 ± 0.28	−0.55 ± 0.28	0.34 [0.00; 0.67] -/**0.048 ***	−0.37 ± 0.28	0.16 [−0.177; 0.492] -/0.503
MEF75 z-score	−0.13 (−1.45; 0.42)	0.12 (−1.11; 0.82)	−0.24 0.134/0.267	−0.23 (−1.37; 0.38)	0.10 0.572/0.572
**Lung-clearance index**					
LCI 2.5	Group A **: 12.20 ± 0.89	11.60 ± 0.89	0.51 [−0.27; 1.29] 0.169/0.169	9.87 ± 0.61	−0.69 [−1.27; −0.11] **0.014 ***/**0.027 ***
	Group B **: 9.19 ± 0.61				
LCI 2.5 z-score	Group A **: 10.1 (7.7; 13.4)	9.40 (6.65; 12.60)	0.07 0.365/0.365	7.10 (4.35; 9.35)	−0.80 **0.007 ***/**0.014 ***
	Group B **: 6.3 (2.62; 8.9)				
**Body plethysmography**					
RV z-score	Group A **: 1.64 (0.31; 2.55)	1.42 (0.14; 2.58)	0.22 0.126/0.252	0.17 (−0.46; 1.14)	−0.05 0.587/0.587
	Group B **: 0.12 (−0.99; 0.86)				
TLC z-score	0.25 (−1.46; 0.77)	0.05 (−0.83; 1.06)	0.20 0.620/1.000	−0.26 (−0.98; 0.88)	0.51 0.558/1.000
RV/TLC z-score	1.10 ± 0.37	1.21 ± 0.51	−0.11 [−1.46; 2.12] -/0.979	0.33 ± 0.51	0.77 [−0.58; 0.50] -/0.356
FRC z-score	0.55 ± 0.34	0.38 ± 0.34	0.17 [−0.41; 0.75] -/0.753	0.50 ± 0.34	0.05 [−0.53; 0.63] -/0.974
Reff z-score	1.04 (−0.38; 3.16)	0.86 (0.01; 2.06)	0.19 0.715/0.779	0.79 (−0.39; 2.11)	0.26 0.390/0.779
sReff z-score	2.78 ± 0.65	2.62 ± 0.65	0.16 [−0.66; 0.97] -/0.890	2.61 ± 0.65	0.16 [−0.65; 0.98] -/0.880
Rtot z-score	1.31 ± 0.09 *(in log scale)*	1.33 ± 0.09 *(in log scale)*	−0.01 [−0.18; 0.16] -/0.979	1.30 ± 0.09 *(in log scale)*	0.01 [−0.16; 0.18] -/0.991

Legends: forced expiratory volume in 1 s (FEV1), forced vital capacity (FVC), maximal expiratory flow (MEF), lung-clearance index (LCI2.5), airway resistance (sReff, kPa*s), airway resistance (Rtot, kPa/(l/s)), airway resistance (Reff, kPa/(l/s)), functional residual capacity (MBNW), total lung capacity (TLC), residual volume (RV), central lung resistance (R5hz), peripheral lung resistance (R5-20hz), peripheral lung reactance (X5hz), peripheral lung reactance (X5hz), area of reactance (AX), and resonant frequency (Fres). *p*-value in bold with * indicates *p* < 0.05; ** baseline includes baseline data of all patients (group A and B) except when there was a significant group effect or a significant interaction group x treatment effect in the statistical model. In this case, group A baseline data were used for the comparison with the first series of Simeox^®^ values (visit 2), and group B baseline data were used for the comparison with the first series of conventional treatment values (visit 2); ^§^ *p*-values were corrected using Tukey (parametric models) or Holm (non-parametric models) methods.

**Table 6 children-10-00204-t006:** Comparisons CFQ-R treatments against baseline.

Cystic Fibrosis Questionnaire (CFQ-R)	Baseline **	Simeox^®^ Treatment		Conventional Treatment	
		After Treatment	Baseline–Treatment	After Treatment	Baseline–Treatment
	Adjusted Means ± SE or Medians (Q1; Q3)	Adjusted Means ± SE or Medians (Q1; Q3)	Treatment Effect [CI95%] Uncorrected/Corrected ^§^ *p*-Value	Adjusted Means ± SE or Medians (Q1; Q3)	Treatment Effect [CI95%] Uncorrected/Corrected ^§^ *p*-Value
Physical score (patient)	85.3 ± 2.1	90.4 ± 2.1	−5.10 [−9.09; −1.12] -/**0.009 ***	87.0 ± 2.1	−1.74 [−5.73; 2.25] -/0.554
Respiratory score (patient)	80.4 ± 2.01	82.7 ± 2.01	−2.29 [−6.52; 1.93] -/0.402	82.5 ± 2.01	−2.08 [−6.31; 2.14] -/0.470

Legends: *p*-value in bold with * indicates *p* < 0.05; ** baseline includes baseline data of all patients (group A and B).

## Data Availability

The data presented in this study are available on request from the corresponding author. The data are not publicly available due to privacy restrictions.

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
