# Peer review of "Efficacy of the Simeox® Airway Clearance Technology in the Homecare Treatment of Children with Clinically Stable Cystic Fibrosis: A Randomized Controlled Trial"

_children, 2023, doi:10.3390/children10020204_

Round 1
Reviewer 1 Report
This is a small study investigating the effect of the airway clearance device Simeox in 40 patients with CF in a cross-over design. Although this is an interesting study, there are some questions concerning the statistics and the interpretation of the results. Further, limitations of the study should be adequately addressed and discussed.
1. Correction for multiple testing was performed using FDR or Benjamini–Hochberg. However, uncorrected p-values are used in the results and to show treatment effects. I suggest to only state trends if the corrected p-value is not significant.
2. I suggest moving the description of the intervention from the online supplement to the main manuscript.
3. The paragraph with the general discussion should be moved to the introduction and shortened substantially as it does not discuss the study results.
4. The discussion on the patient population could be moved to the online supplement.
5. How many patients were on dual or triple CFTR modulator in this study? The paragraph on the additional effect of the Simeox device on CFTR modulators seems highly speculative, since it is not described that the additional effect was studied here. In general, the discussion should focus more on the study results and the comparison the other therapeutic options.
6. line 24 and Line 366. There is no evidence by the results that Simeox therapy improves the drainage of distal airways.
7. Limitations should be addressed in the main manuscript. Especially as ths study was performed in a cross over design and during the start of COVID pandemic. How was the study and the results influenced by the pandemic?
Minor comments:
Do the tables 2 and 3 display the change from baseline or only the value at the end of the intervention. Please clarify and describe in the table heading or legend.
References to Figures and tables need to be updated
Author Response
- Correction for multiple testing was performed using FDR or Benjamini–Hochberg. However, uncorrected p-values are used in the results and to show treatment effects. I suggest to only state trends if the corrected p-value is not significant.
Corrected.
- I suggest moving the description of the intervention from the online supplement to the main manuscript.
Corrected.
- The paragraph with the general discussion should be moved to the introduction and shortened substantially as it does not discuss the study results.
Corrected.
- The discussion on the patient population could be moved to the online supplement.
Corrected.
- How many patients were on dual or triple CFTR modulator in this study? The paragraph on the additional effect of the Simeox device on CFTR modulators seems highly speculative, since it is not described that the additional effect was studied here. In general, the discussion should focus more on the study results and the comparison the other therapeutic options.
There were no patients on CFTR modulator therapy in this study. Therefore, we decided to delete the paragraph mentioned.
- line 24 and Line 366. There is no evidence by the results that Simeox therapy improves the drainage of distal airways.
We replaced „distal Airways” by „the airways”.
- Limitations should be addressed in the main manuscript. Especially as ths study was performed in a cross over design and during the start of COVID pandemic. How was the study and the results influenced by the pandemic?
We moved the limitations paragraph to the main manuscript. The pandemic did not affect significantly the study and the results.
Minor comments:
Do the tables 2 and 3 display the change from baseline or only the value at the end of the intervention. Please clarify and describe in the table heading or legend.
Clarified and corrected.
References to Figures and tables need to be updated.
It was updated.
Reviewer 2 Report
The manuscript “Efficacy of the Simeox® Airway Clearance Technology in the Homecare Treatment of Children With Clinically Stable Cystic Fibrosis: A Randomized Controlled trial” describes the clinical trial of a new ACT (Simeox) in 40 pediatric CF patients with stable disease. The study is well designed and the results clearly described. Of course, the number of patients is low, but CF is a rare disease and the trial valid under the circumstances. However, there are some concerns with need to be addressed:
1: Please explain all acronyms and expressions.
Abstract: improvement in R20Hz and MEF75%, CFQ-R physical score: expected to know these terms, but for people not familiar with the terms, they should be explained at first mention.
Line 260: Please explain the acronym O-PEP.
2: You write in the conclusions of the abstract: Simeox may improve drainage of proximal and distal airway. This is in contrast to line 307, where you write: thus we suggest that airway obstruction in proximal airways was decreased in Simeox®. Please explain.
3: You write on line 64 that “the drainage is performed mainly in the distal parts of the bronchial tree, which is very important in obstructive respiratory diseases (in CF in particular) where the smallest caliber of the bronchi is most affected. The reference given is 10, which is given as: “drainage and the active cycle of breathing techniques with postural drainage. Thorax, 50:165-169”.
A. Is this the reference to support your claim that drainage is performed mainly in the distal parts?
B. Please give a correct reference.
C. This does not conform with your own results that the effect was in proximal airways. Please comment on this discrepancy.
Author Response
1: Please explain all acronyms and expressions.
Abstract: improvement in R20Hz and MEF75%, CFQ-R physical score: expected to know these terms, but for people not familiar with the terms, they should be explained at first mention.
Line 260: Please explain the acronym O-PEP.
Corrected.
2: You write in the conclusions of the abstract: Simeox may improve drainage of proximal and distal airway. This is in contrast to line 307, where you write: thus we suggest that airway obstruction in proximal airways was decreased in Simeox®. Please explain.
We changed our conclusions. We replaced „distal Airways” by „the airways”.
3: You write on line 64 that “the drainage is performed mainly in the distal parts of the bronchial tree, which is very important in obstructive respiratory diseases (in CF in particular) where the smallest caliber of the bronchi is most affected. The reference given is 10, which is given as: “drainage and the active cycle of breathing techniques with postural drainage. Thorax, 50:165-169”.
- Is this the reference to support your claim that drainage is performed mainly in the distal parts?
- Please give a correct reference.
- This does not conform with your own results that the effect was in proximal airways. Please comment on this discrepancy.
There was a mistake in reference number. We gave the correct reference.